# Interconnectedness of Cryptocurrency Uncertainty Indices with Returns and Volatility in Financial Assets during COVID-19

**Awad Asiri** [1], **Mohammed Alnemer** [2] **and M. Ishaq Bhatti** [3,4,*]

1 College of Business Administration, Jazan University, Jizan 45142, Saudi Arabia; aasiri@jazanu.edu.sa
2 College of Business Administration, Shaqra University, Shaqra 15526, Saudi Arabia; malnemer@su.edu.sa
3 UBD School of Business and Economics, Universiti Brunie Darussalam, Gadong BE 1410, Brunei
4 Brunei & La Trobe Business School, La Trobe University, Melbourne, VIC 3086, Australia
* Correspondence: i.bhatti@latrobe.edu.au

**Abstract:** This paper investigates the dynamic relationship between cryptocurrency uncertainty indices and the movements in returns and volatility across spectrum of financial assets, comprising cryptocurrencies, precious metals, green bonds, and soft commodities. It employs a Time-Varying Parameter Vector Autoregressive (TVP-VAR) connectedness approach; the analysis covers both the entire sample period spanning August 2015 to 31 December 2021 and the distinct phase of COVID-19 pandemic. The findings of the study reveal the interconnectedness of returns within these asset classes during the COVID-19 pandemic. In this context, cryptocurrency uncertainty indices emerge as influential transmitters of shocks to other financial asset categories and it significantly escalates throughout the crisis period. Additionally, the outcomes of the study imply that during times of heightened uncertainty, exemplified by events such as the COVID-19 pandemic, the feasibility of portfolio diversification for investors might be constrained. Consequently, the amplified linkages between financial assets through both forward and backward connections could potentially compromise financial stability. This research sheds light on the impact of cryptocurrency uncertainty on the broader financial market, particularly during periods of crisis. The findings have implications for investors and policymakers, emphasizing the need for a comprehensive understanding of the interconnectedness of financial assets and the potential risks associated with increased interdependence. By recognizing these dynamics, stakeholders can make informed decisions to enhance financial stability and manage portfolio risk effectively.

**Keywords:** COVID-19 pandemic; dynamic connectedness; TVP-VAR model; precious metals

**JEL Classification:** C22; D81; G15

## 1. Introduction

In recent years, research related to cryptocurrency investment has generated significant debate within the financial sphere. As these digital assets are continuing to grow and gain recognition, comprehending their dynamics and influence on the broader financial landscape is important. Compounding this complexity, the advent of the COVID-19 pandemic and Russia–Ukraine conflict has introduced an additional layer of uncertainty into global financial markets. This study aims to investigate into the interplay between cryptocurrency uncertainty indices and the dynamic movement of returns and volatility in various financial assets during the COVID-19 crisis. Through this paper, we attempt to examine the interdependence among cryptocurrency uncertainty indices, precious metals, green bonds, and soft commodities. Our aim is to investigate the transmission channels and potential spillover effects during times of heightened uncertainty. The findings derived from this study will contribute to the existing literature by fostering a deeper comprehension of the interconnectedness inherent in financial assets. Furthermore, these findings will

provide insights to investors, policymakers, and financial institutions with tools to manage risks and keep financial stability.

The volatility in financial markets is an important factor that plays a significant role in economics, serving as a pivotal risk indicator. A nuanced understanding of the drivers behind volatility across various financial assets holds great importance for stakeholders including academics, investors, regulators, and speculators. This understanding helps in assessing the potential risks that could undermine the stability of the financial system. Stakeholders closely monitor the propagation of both volatility and returns across diverse assets and markets. Although the correlation between returns and volatility has been extensively examined across different financial markets in the existing literature, the spillover effects of cryptocurrencies and their uncertainty on other financial assets have received limited scholarly attention within academia and the financiers. Therefore, there is a need to explore the linkages and interdependencies between cryptocurrency uncertainty indices and the volatility of other financial assets. By doing so, we can foster a deeper comprehension of the intricate dynamics at play within these markets.

The cryptocurrency uncertainty indices play a vital role in enabling investors to gauge uncertainty within the cryptocurrency market, an aspect not fully captured by conventional uncertainty measures such as economic policy uncertainty (Al-Yahyaee et al. 2019; Antonakakis et al. 2013; Demir et al. 2018), VIX volatility index (Alqahtani and Chevallier 2020; Fakhfekh et al. 2021), Investor Attention Index (Smales 2022), and Twitter Economic Uncertainty index (El Khoury and Alshater 2022; Gök et al. 2022). These indices often fall short in accurately reflecting the surge in uncertainty within the cryptocurrency market, a pivotal determinant of asset returns. Notably, heightened uncertainty within the cryptocurrency market directly influences investor returns (Bashir and Kumar 2023). Thus, cryptocurrency uncertainty indices act as instruments that highlight the shift in the cryptocurrency market in response to various events, such as COVID-19 pandemic (Khan et al. 2023). This research sheds light on the impact of cryptocurrency uncertainty on the broader financial market, particularly during periods of crisis. The findings have implications for investors and policymakers, emphasizing the need for a comprehensive understanding of the interconnectedness of financial assets and the potential risks associated with increased interdependence. By recognizing these dynamics, stakeholders can make informed decisions to enhance financial stability and manage portfolio risk effectively.

Nonetheless, the role of cryptocurrencies in the global financial system is increasing every day as this is an important investment asset for most retail and institutional investors. The total market capitalization of cryptocurrencies surpassed USD 1.29T in May 2022. Bitcoin leads the market with a market capitalization of USD 577B and 44% share of the cryptocurrency market. In recent times, Bitcoin's dominance has fallen with the rise in stable coins (Ghabri et al. 2022; Kristoufek 2021; Wang et al. 2020), asset-backed cryptocurrencies (Aloui et al. 2021; Jalan et al. 2021; Yousaf and Yarovaya 2022a), decentralized finance assets (DeFi) (Yousaf et al. 2022; Yousaf and Yarovaya 2022b), and non-fungible tokens (NFTs) (Aharon and Demir 2021; Yousaf and Yarovaya 2022b). Cryptocurrencies exhibited higher volatility in the global COVID-19 crisis, which also affected the cryptocurrency market. The World Health Organization (WHO) declared COVID-19 as global pandemic on 11 March 2020; after this announcement, the Bitcoin price was $3953 on 11 March, and it sharply rose during the pandemic as retail and institutional investors shifted their investments from equity markets to cryptocurrencies and other non-traditional financial assets due to the safe-haven role of cryptocurrencies (Bouri et al. 2020; Corbet et al. 2020b; Rubbaniy et al. 2021a).

The cryptocurrency market is highly volatile; many investors want to invest in the market in the hope of getting higher returns during financial turmoil. During turbulent periods, regulators, policymakers, and investors are interested in observing the return and volatility spillovers for: firstly, decisions about portfolio diversification; and secondly, implementing policies for financial stability. These issues are relevant to the COVID-19 pandemic, when the unemployment rate increased, halting economic activities as economic uncertainty results in financial chaos that disturbed the portfolio asset allocations and

reduced financial stability. The COVID-19 pandemic hugely disrupted financial markets and affected all economy sectors, which ultimately triggered the global recession. With the increase in systematic risk during the COVID-19 outbreak, market participants were interested in obtaining information about volatility transmission among various financial assets for portfolio diversification. Investors re-balanced their portfolios during the financial turmoil by switching from risky to safe-haven assets (Bouri et al. 2021b; Choudhry et al. 2015; Ha and Dai 2022; Khan et al. 2023; Ghouse et al. 2023).

The existing literature largely ignored the interaction of cryptocurrencies and their interaction with other relatively safe traditional financial assets. The COVID-19 pandemic also changed the co-movements between cryptocurrencies and traditional assets. Hence, this study focuses on uncovering the drivers of cryptocurrencies and traditional asset return and volatility spillovers as information transmission among financial markets is extensively studied (Bação et al. 2018; Forbes and Rigobon 2002; Kurka 2019). Existing studies also discussed the connectedness of financial assets during the financial crisis. The important works of Diebold and Yilmaz (2009, 2012, 2014) developed a quantitative measure of dynamic connectedness based on forecast error variance decomposition using VAR models. We try to contribute to the relevant literature by investigating dynamic connectedness of different assets and cryptocurrency uncertainty indices. Specifically, we are interested in examining the dynamic connectedness during the COVID-19 shock by considering its time-varying structure. Thus, we try to answer the following questions: What role has the COVID-19 pandemic played in exhibiting the return and cryptocurrency uncertainty connectedness of different financial assets? Are the asset returns time-varying in nature? Do the cryptocurrency uncertainty indices explain the return and volatility connectedness among financial assets?

We used indices for cryptocurrencies, precious metals, green bonds, and soft commodities and cryptocurrency uncertainty to apply the time-varying parameter vector auto-regressions (TVP-VAR) dynamic connectedness approach to answer the above research questions. To the best of our knowledge, this is the first study of its kind on the dynamic connectedness of returns and volatility of different assets during a financial crisis (Adekoya and Oliyide 2021; Bouri et al. 2021a; Corbet et al. 2020a; Kamal and Hassan 2022; So et al. 2020). This paper also investigates the response of financial assets to the COVID-19 pandemic by extending published studies conducted in different financial markets (Adekoya and Oliyide 2021; Baig et al. 2020; Le et al. 2021b; Rubbaniy et al. 2021b). The usage of the TVP-VAR approach by Antonakakis et al. (2020) overcomes the shortcomings (e.g., outlier sensitivity, short time, rolling window size) of the original connectedness approach by Diebold and Yilmaz (2009, 2012, 2014). The TVP-VAR approach also serves to measure cross-asset connectedness in the network.

Furthermore, existing studies have discussed the role of various uncertainty indices in shaping the dynamics of cryptocurrency returns and volatility. These different measures of uncertainty encompass the economic policy uncertainty index (Elsayed et al. 2022a; Foglia and Dai 2021; Yen and Cheng 2021), Twitter-based uncertainty index (Aharon et al. 2022; Wu et al. 2021), and the economic and political uncertainty (Colon et al. 2021; Kyriazis 2021) and cryptocurrency uncertainty indices (Elsayed et al. 2022b; Lucey et al. 2022).

In a recent study, Yousaf and Goodell (2023) investigated the central banks' digital currencies (CDBC), cryptocurrency policy uncertainty index as well as digital payments stocks by using the dynamic connectedness approach. Their findings highlight the transmission of shocks from UCRY policy and price to digital payment stocks. Moreover, they identified the limited interconnectedness between cryptocurrency uncertainty indices and digital payment stocks, indicating their potential hedging tools against cryptocurrency market volatility. Yan et al. (2022) investigated the impact of cryptocurrency uncertainties on sustainable and traditional mutual fund and found that traditional mutual funds' investments are influenced by uncertainty in the cryptocurrencies market.

Wei et al. (2023) delved into safe-haven properties of cryptocurrencies and forecasting ability of cryptocurrency uncertainty indices for volatility in precious metals. Employ-

ing the GARCH-MIDAS approach, their results show that the forecasting prowess of cryptocurrency uncertainty indices in the precious metals market.

In our contribution to the field, we leverage novel cryptocurrency uncertainty indices in conjunction with various asset classes—namely cryptocurrencies, precious metals, green bonds, and soft commodities. Furthermore, we extend the work initiated by Elsayed et al. (2022b) on dynamic connectedness between gold, cryptocurrency index and cryptocurrency uncertainty indices. Our findings show the higher returns and volatility connectedness in the overall sample and during the COVID-19 pandemic, and most financial assets are net receivers of shocks. The patterns of return and volatility spillover changed during the pandemic for most financial assets. Overall, our findings suggest that cryptocurrency uncertainty indices and transmitters of shocks extend to other financial assets. The COVID-19 pandemic resulted in a spike of risk in financial markets and the magnitude of dynamic connectedness increased during the first wave of COVID-19 which is like the finding of Bhatti and Ghouse (2023). Hence, risk-averse equity market investors can minimize such risks by investing in less-connected assets to diversify portfolios. The remainder of the study is as follows. Section 2 presents the relevant literature review. In Section 3, we describe the methodology and data. In Section 4, we show the findings of the study. Finally, Section 5 concludes the study.

## 2. Literature Review

Studies have discussed the return and volatility transmission across financial assets using different methods, for instance Granger causality (Adekoya and Oliyide 2021; Albulescu et al. 2019; Zhang and Broadstock 2020) and dynamic conditional correlation (Abuzayed and Al-Fayoumi 2021; Hassan et al. 2019). However, the existing literature highlights the usefulness of connectedness of financial assets using the dynamic connectedness approach (Shahzad et al. 2021a, 2021b). The higher inter-connectedness among financial assets indicates greater market risk, and investors minimize their risks by investing in weakly connected financial assets. The higher market risk in the network explains the instability of the financial markets. Dynamic return connectedness is used to identify the isolated assets so that these assets function as hedge or safe haven against the risk of other financial assets.

The literature on return and volatility connectedness among different financial assets, such as equity, bonds, and commodities, are scarce. Some authors studied the link between commodities, currency, and equity markets. For instance, Kang et al. (2017) studied the price transmission among crude oil, agricultural commodities and precious metals using the DECO-GARCH model. They detected an increase in spillover impacts during the financial crisis. Lundgren et al. (2018) also tested the connectedness and causality by using equities, currencies, oil, and US treasury bonds, as well as different proxies of uncertainty (EU and US EPU and VIX) using data from 2004–2016, and they found the uncertainty proxies were net transmitters of shocks during the financial crisis.

Mensi et al. (2017) investigated the spillovers between gold, Dow Jones, conventional, Islamic, technology, financial, and telecommunications sector and sustainable indices. These authors found that gold, energy, technology and telecom sectors and receivers of shocks and Dow Jones indices contribute to risk spillovers. Yoon et al. (2019) investigated dynamic and static returns connectedness among equity, bond, commodity, and currency markets. They identified the Shanghai stock exchange, Nikkei 225, and KOSPI are receivers of spillover shocks. Kumar et al. (2019) investigated volatility and correlation between stock prices, natural gas, and oil in India via the VARMA-DCC GARCH models. Their findings highlights highest short-term spillovers between oil and natural gas. Iglesias-Casal et al. (2020) discussed the volatility spillovers and diversification potential of oil, gold and clean energy indices in Brazil by using BEKK and A-DCC models. They emphasized gold's higher diversification potential and optimal portfolio weights.

A recent study by Mensi et al. (2020) explored the risk spillovers between energy futures and precious metals, noting increased volatility spillovers during the financial crisis. They observed that gold and oil transmit volatility to other financial assets.

Bouri et al. (2021a) explored return connectedness with crude oil, equities, currencies, and bonds during the COVID-19 pandemic using the TVP-VAR approach. They observed changes in connectedness network's structure and identified equity and USD indices as shock transmitters before COVID-19, while bond indices become the volatility shock transmitter during the outbreak.

Asl et al. (2021) analyzed volatility transmission between clean energy indices and energy commodities using an asymmetric BEKK-MGARCH(1,1) model. They found higher optimal weights and hedging effectiveness for clean energy indices, making them useful for hedging equity risks in the energy sector. They concluded that investors can invest in green assets to hedge the equity risk for stocks in energy sector. Further, Szczepanska-Przekota (2021) explored the impact of cryptocurrencies on economic conditions of different markets, and found that investors perceive the cryptocurrency market as more risky as compared to equity markets.

Shahid et al. (2023) explored the interconnectedness and risk transmission across global financial markets and assessed the portfolio diversification potential of socially responsible investments using DCC-GARCH and VAR-GARCH models. Their findings indicated negative correlation between traditional volatility indices and socially responsible investment indices. They also found that implied volatility indices of silver and golds hedge the risks against SRIs investments. Furthermore, Elsayed et al. (2022b) extended the above research by examining the return and volatility spillovers in gold, cryptocurrency index, and cryptocurrency price and policy uncertainty indices. They found that cryptocurrency policy uncertainty is the transmitter of shocks to other assets while gold is the receiver of shocks. We extend the above research by investigating the dynamic returns and volatility spillovers among various financial assets and cryptocurrency uncertainty indices developed by Lucey et al. (2022). The news-based uncertainty indices are relevant to cryptocurrencies and can better predict uncertainty in the cryptocurrency market.

## 3. Research Methodology

### 3.1. The Data

To test and study the dynamic connectedness of different financial asset returns and cryptocurrency price and policy uncertainty index constructed by Lucey et al. (2022), we collected the weekly data of cryptocurrency uncertainty indices from the authors' website[1]. The cryptocurrency uncertainty indices were constructed using news articles related to cryptocurrency on the Lexis Nexis database. We also gathered the daily closing price of Bitcoin and Ethereum from the Coin Market Cap website[2]. The closing price data for precious metals (gold, silver, platinum), S&P green bonds, and S&P GSCI soft commodities index were downloaded from the DataStream database provided by Thomson Reuters[3]. The final sample includes data from 7 August 2015 to 31 December 2021.

In the next step, we converted the daily closing prices of financial assets into log returns that were further converted into weekly returns to estimate dynamic returns and volatility connectedness at a weekly frequency. The dynamic connectedness requires that all series follow non-stationary unit root test processes. Hence, for implementing the dynamic connectedness approach, we transformed the data using the first log-difference of series: $y_{it} = log(x_{it}) - log(x_{it} - 1)$. The selected financial assets are relatively stable during the period of extreme volatility (Le et al. 2021b; Mo et al. 2022; Su et al. 2022; Umar et al. 2021) and essential for the stability of financial markets due to their volatility to other markets. Hence, it would be interesting to study the connection between these financial assets with cryptocurrency uncertainty. In addition, well, our dataset includes data for the period of the COVID-19 pandemic, which is useful to observe the returns and volatility connectedness during it. We offer a snapshot of the data in the following Table 1, which includes the

descriptive statistics of cryptocurrency indices, cryptocurrencies, precious metals, green bonds, and soft commodities indices.

**Table 1.** Descriptive Statistics of Sample.

| | | | | | | | | | |
|---|---|---|---|---|---|---|---|---|---|
| **Panel A: Descriptive Statistics Full Sample (7 August 2015 to 31 December 2021).** | | | | | | | | | |
| Variables | UCRY_Policy | UCRY Price | Bitcoin | Ethereum | Gold | Silver | Platinum | SP Green Bonds | SP GSCI Softs |
| Mean | 0.000 | 0.000 | 0.024 | 0.058 | 0.001 | 0.003 | 0.003 | 0.000 | 0.001 |
| Variance | 0.000 | 0.000 | 0.002 | 0.018 | 0.000 | 0.000 | 0.000 | 0.000 | 0.000 |
| Skewness | 8.081 *** | 6.617 *** | 4.795 *** | 5.461 *** | 4.930 *** | 13.001 *** | 10.276 *** | 9.971 *** | 4.146 *** |
| | 0.000 | 0.000 | 0.000 | 0.000 | 0.000 | 0.000 | 0.000 | 0.000 | 0.000 |
| Ex.Kurtosis | 76.285 *** | 53.141 *** | 29.936 *** | 37.572 *** | 36.607 *** | 193.579 *** | 120.734 *** | 114.121 *** | 28.471 *** |
| | 0.000 | 0.000 | 0.000 | 0.000 | 0.000 | 0.000 | 0.000 | 0.000 | 0.000 |
| JB | 84,622.361 *** | 41,738.229 *** | 13,751.343 *** | 21,305.339 *** | 20,002.199 *** | 530,908.103 *** | 208,736.085 *** | 186,779.394 *** | 12,237.996 *** |
| | 0.000 | 0.000 | 0.000 | 0.000 | 0.000 | 0.000 | 0.000 | 0.000 | 0.000 |
| ERS | −5.437 *** | −4.707 *** | −6.351 *** | −0.871 | −6.624 *** | −5.640 *** | −6.281 *** | −4.318 *** | −4.953 *** |
| | 0.000 | 0.000 | 0.000 | 0.000 | 0.000 | 0.000 | 0.000 | 0.000 | 0.000 |
| Q(10) | 91.739 *** | 126.632 *** | 45.933 *** | 63.807 *** | 52.338 *** | 52.491 *** | 89.844 *** | 124.134 *** | 22.726 *** |
| | 0.000 | 0.000 | 0.000 | 0.000 | 0.000 | 0.000 | 0.000 | 0.000 | 0.000 |
| Q2(10) | 15.345 *** | 8.362 | 3.844 | 35.637 *** | 13.855 *** | 12.043 ** | 65.188 *** | 70.642 *** | 7.058 |
| | −0.005 | −0.146 | −0.69 | 0 | −0.01 | −0.026 | 0 | 0 | −0.248 |
| **Panel B: Descriptive Statistics Full Sample COVID-19 (1 January 2020 to 31 December 2021)** | | | | | | | | | |
| Mean | 0.000 | 0.000 | 0.022 | 0.041 | 0.001 | 0.005 | 0.005 | 0.000 | 0.002 |
| Variance | 0.000 | 0.000 | 0.002 | 0.006 | 0.000 | 0.000 | 0.000 | 0.000 | 0.000 |
| Skewness | 4.606 *** | 3.574 *** | 7.074 *** | 4.594 *** | 4.417 *** | 7.727 *** | 5.964 *** | 6.089 *** | 3.909 *** |
| | 0.000 | 0.000 | 0.000 | 0.000 | 0.000 | 0.000 | 0.000 | 0.000 | 0.000 |
| Ex.Kurtosis | 23.186 *** | 14.875 *** | 59.241 *** | 26.666 *** | 26.720 *** | 64.121 *** | 38.122 *** | 38.670 *** | 20.649 *** |
| | 0.000 | 0.000 | 0.000 | 0.000 | 0.000 | 0.000 | 0.000 | 0.000 | 0.000 |
| JB | 2697.329 *** | 1180.259 *** | 16,075.460 *** | 3447.105 *** | 3432.009 *** | 18,851.366 *** | 6914.146 *** | 7122.527 *** | 2112.400 *** |
| | 0.000 | 0.000 | 0.000 | 0.000 | 0.000 | 0.000 | 0.000 | 0.000 | 0.000 |
| ERS | −3.596 *** | −2.898 *** | −3.750 *** | −3.743 *** | −3.391 *** | −4.079 *** | −3.402 *** | −4.030 *** | −3.774 *** |
| | −0.001 | −0.005 | 0.000 | 0.000 | −0.001 | 0.0000 | −0.001 | 0.000 | 0.000 |
| Q (10) | 17.926 *** | 19.297 *** | 10.909 ** | 11.869 ** | 25.382 *** | 16.323 *** | 26.846 *** | 42.238 *** | 13.134 ** |
| | −0.001 | −0.001 | −0.045 | −0.028 | 0 | −0.003 | 0.000 | 0.000 | −0.015 |
| Q2 (10) | 4.088 | 1.439 | 0.644 | 2.008 | 5.521 | 3.597 | 19.948 *** | 21.665 *** | 2.264 |
| | −0.651 | −0.977 | −0.998 | −0.937 | −0.43 | −0.73 | 0.000 | 0.000 | −0.912 |

**Notes:** The symbols ***, ** indicate significance at the 1%, 5% levels; the D'Agostino (1970) skewness test, Anscombe and Glynn (1983) kurtosis test, Jarque and Bera (1980) normality test, Elliott et al. (1996) ERS unit-root test, *Q* (10) & *Q*2(10), and Fisher and Gallagher (2012) weighted portmanteau tests are applied to the dataset.

Table 1 panel A includes the descriptive statistics of the whole sample (7 August 2015, to 31 December 2021) and panel B includes the data covering the COVID-19 pandemic (1 January 2020 to 31 December 2021). In panel A of Table 1, the full sample results show that average returns are positive for most of the series. From the selected financial assets, Bitcoin provides higher returns with a value of 0.058 and, unit-root test processes during the COVID-19 pandemic, gold, silver, platinum, and S&P GSCI soft commodities are increased. Gold returns remain stable in the overall sample during the COVID-19 pandemic. The returns of cryptocurrencies are reduced during the crisis with the average values of Bitcoin and Ethereum being 0.024 and 0.058, respectively, in the overall sample compared to average values of 0.022 and 0.041 during COVID-19. Overall, the weekly returns of these financial assets are not negative and provide better returns to investors in cryptocurrencies and commodities markets in the presence of cryptocurrencies uncertainty.

The difference between Bitcoin and Ethereum is higher in the full sample and during COVID-19, which shows that cryptocurrencies are riskier than precious metals, green bonds, and soft commodities. Further, the positive and significant rightward skewed returns series show that the mean is higher than median in different financial assets used in this study. The kurtosis and Jarque–Bera normality tests confirm that all returns series have fat tails and follow the leptokurtic distribution. Results support the non-normality of the data in line with Jarque and Bera (1980), in which they show that all financial assets

are not normally distributed. The stationarity is tested utilizing the Elliott et al. (1996) ERS unit root test, which shows that all returns series are significant and stationary at the 1% level of significance. Finally, we also checked the goodness-of-fit of financial time series using the Fisher and Gallagher (2012) weighted portmanteau test that is significant at 1% in most cases. It shows the autocorrelation between returns and squared returns is useful for examining the interconnectedness of these financial assets using the TVP-VAR dynamic connectedness approach.

The weekly log returns on financial assets and cryptocurrency uncertainty indices are displayed in Figure 1 below. We take the natural log by following (Demir et al. 2018; Hasan et al. 2021; Xu et al. 2023). Shown here is that prices of cryptocurrencies, precious metals, green bonds, and soft commodities indices show a sharp reduction during the first phase of the COVID-19 pandemic. Furthermore, the cryptocurrency policy and price uncertainty indices rapidly increased as the COVID-19 crisis progressed.

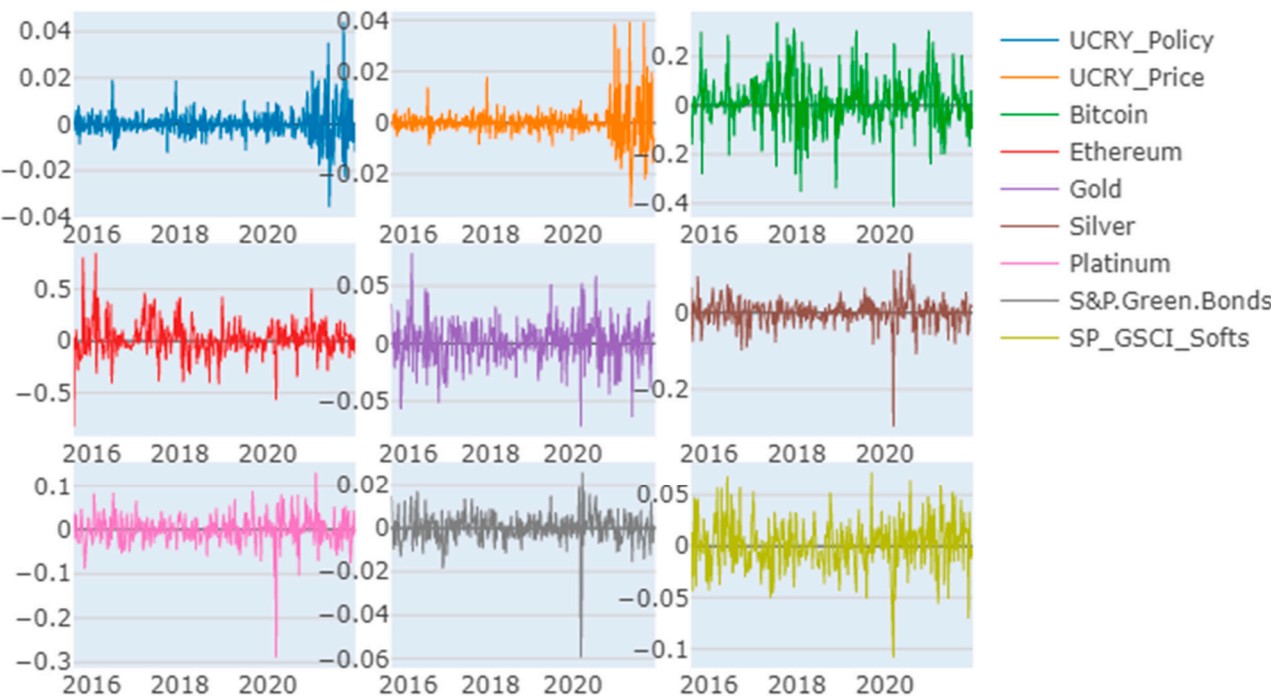

**Figure 1.** Log returns data of cryptocurrencies, precious metals, green bonds, and soft commodities. **Notes:** Figure 1 displays the weekly log returns of cryptocurrencies, precious metals, green bonds, and soft commodities.

### 3.2. The Model

We investigated the impact of cryptocurrency uncertainty indices on the return and volatility connectedness with the Time-Varying Parameter Autoregressive (TVP-VAR) dynamic connectedness approach developed by Antonakakis et al. (2020). Their approach is essentially an extension of Diebold and Yilmaz (2009, 2012, 2014). Selecting this econometric method is based on existing studies for testing the dynamic connectedness among financial markets. This method is useful when correlation among financial markets rises during times of financial turmoil. The dynamic conditional correlation models provide biased results during the crisis as they are based on market state and macroeconomic factors, yet the real connectedness among markets is not affected by the financial turbulence. In contrast, the spillover approach is based on Forecast Error Variance Decomposition (FEVD), which originated from the Vector Autoregressive model (VAR) model and not affected by the conditional correlation (Elsayed et al. 2022b; Elsayed and Helmi 2021; Umar et al. 2022).

The spillover approach devised by Diebold and Yilmaz (2009, 2012, 2014) is used to obtain the information about shock in one variable to another variable based on FEVD.

However, the spillover method has several drawbacks: first, the arbitrary selection of rolling-window size is not useful in small samples; second, original connectedness approach by Diebold and Yilmaz (2009, 2012, 2014) is highly sensitivity to outliers and based on normalization techniques, which may provide biased results (Caloia et al. 2019); and third, the original dynamic connectedness approach may produce biased estimations due to sign and rank errors. These issues are solved using the TVP-VAR dynamic connectedness approach devised by Antonakakis et al. (2020). The window size is ideally estimated by mean squared prediction errors based on the multivariate Kalman Filters (Koop and Korobilis 2013, 2014). This technique is also useful for dealing with outliers, especially during the financial crisis. The short time series, and sign and rank errors are also reduced in the TVP-VAR approach with scaler-based normalization of Generalized Forecast Error Variance.

The return and volatility spillovers are discussed in existing literature using the original dynamic connectedness approach (Diebold and Yilmaz 2009, 2012, 2014), TVP-VAR approach (Adekoya and Oliyide 2021; Bouri et al. 2021a; Dai et al. 2022; Elsayed et al. 2022b) and extended joint connectedness based on TVP-VAR (Balcilar et al. 2021; Chen et al. 2022). Meanwhile the TVP-VAR approach is based on auto-regressive conditional heteroscedasticity (ARCH) process proposed by Engle (1982), and the generalized autoregressive conditional heteroscedasticity (GARCH) approach by Bollerslev (1986) to overcome problems in ARCH models. Consequently, the TVP-VAR approach of Antonakakis et al. (2020) examines the return and volatility transmission across cryptocurrencies, precious metals, green bonds, soft commodities, and cryptocurrency uncertainty indices.

*3.3. Research Methods*

3.3.1. Time-Varying Parameter Vector Autoregression (TVP-VAR)

The TVP-VAR dynamic connectedness approach of Antonakakis et al. (2020) is applied to assess the dynamic connectedness between various financial assets. The TVP-VAR model with the lag-length of order one is selected by the Bayesian information criterion (BIC), and we choose the rolling window of 52 weeks and weekly returns which are nth-transformed to control the stationarity issues:

$$
\begin{aligned}
y_t &= B_t y_{t-1} + \epsilon_t & \epsilon_t &\sim N(0, \Sigma_t) \\
\mathrm{vec}(B_t) &= \mathrm{vec}(B_{t-1}) + v_t & v_t &\sim N(0, R_t)
\end{aligned}
\tag{1}
$$

where Equation (1) shows that $y_t, y_{t-1}$ and $\epsilon_t$ are $K \times 1$ dimensional vector and $B_t$ and $\Sigma_t$ are $K \times K$ dimensional matrices.

The symbols $\mathrm{vec}(B_t)$ and $v_t$ are $K^2 \times 1$ dimensional vectors, whereas $R_t$ is a $K^2 \times K^2$ dimensional matrix. All parameters $(B_t)$ are allowed to use the TVP-VAR approach, which is also helpful for examining the time-varying relationship and variance-covariance matrices; $\Sigma_t$ and $R_t$.

Further, the Wold theorem is applied to transform the model to the TVP-VMA model in

$$
y_t = \sum_{h=0}^{\infty} A_{h,t} \epsilon_{t-i}
\tag{2}
$$

where Equation (2) shows that $A_0 = I_K$ and $\epsilon_t$ is a vector of white noise symmetric shocks with $K \times K$ time-varying covariance matrix of $E(\epsilon_t \epsilon_t') = \Sigma_t$.

Therefore, in the next step, the $H$-step forecast error is estimated in Equation (3);

$$
\begin{aligned}
\xi_t(H) &= y_{t+H} - E(y_{t+H} \mid y_t, y_{t-1}, \ldots) \\
&= \sum_{h=0}^{H-1} A_{h,t} \epsilon_{t+H-h},
\end{aligned}
\tag{3}
$$

with forecast error covariance matrix equal to in Equation (4):

$$
E\big(\xi_t(H) \xi_t'(H)\big) = A_{h,t} \Sigma_t A_{h,t}'.
\tag{4}
$$

### 3.3.2. The Generalized Dynamic Connectedness Approach

The generalized dynamic connectedness approach is based on the *H*-step ahead of generalized forecast error variance decomposition (GFEVD); $gSOT_{ij,t}$, is also applied and it can be interpreted as the effect the shock in variable *j* has on variable *i*. This process is explained in Equation (5) below:

$$
\begin{aligned}
\zeta_{ij,t}^{gen}(H) &= \frac{E\left(\xi_{i,t}^2(H)\right) - E\left[\xi_{i,t}(H) - E\left(\xi_{i,t}(H)\right)|\epsilon_{j,t+1},\dots,\epsilon_{j,t+H}\right]^2}{E\left(\xi_{it}^2(H)\right)} \\
&= \frac{\sum_{h=0}^{H-1}\left(e_i' A_{ht}\Sigma_t e_j\right)^2}{\left(e_j'\Sigma_t e_j\right)\sum_{h=0}^{H-1}\left(e_i' A_{ht}\Sigma_t A_{ht}' e_i\right)} \\
gSOT_{ij,t} &= \frac{\zeta_{ij,t}^{gen}(H)}{\sum_{j=1}^{K}\zeta_{ij,t}^{gen}(H)}
\end{aligned}
\tag{5}
$$

where $e_i$ is a $K \times 1$ zero selection vector with unity on its *i*th position and $\zeta_{ij,t}^{gen}(H)$ denotes the proportional reduction of the *H*-step forecast error variance of variable *i* due to conditioning on future shocks of variable *j*.

The generalized dynamic spillover average table displays the total connectedness to demonstrate total connectedness among financial assets from shock in one variable to the whole network. This dynamic connectedness metric is explained in below in Equation (6):

$$
\begin{aligned}
S_{i\leftarrow\cdot,t}^{gen,from} &= \sum_{j=1,i\neq j}^{K} gSOT_{ij,t} \\
S_{i\rightarrow 0,t}^{gen,to} &= \sum_{j}^{K} gSOT_{ji,t}
\end{aligned}
\tag{6}
$$

Another measure is the net total directional connectedness of variable *i*, and it displays whether variable *i* influences the network more than being influenced by it and it is presented in Equation (7):

$$
S_{i,t}^{gen,net} = S_{i\rightarrow 0,t}^{gen,to} - S_{i\leftarrow\cdot,t}^{gen,from}
\tag{7}
$$

If $S_{i,t}^{gen,net} > 0 \left(S_{i,t}^{gen,net} < 0\right)$, variable *i* is a net transmitter (receiver) of shocks which shows that variable *i* is driving (driven by) the network.

The next metric is TCI is average total directional connectedness from (to) others and we explain it in Equation (8), which is shown here:

$$
gSOI_t = \frac{1}{K}\sum_{i=1}^{K} S_{i\leftarrow,t}^{gen,from} = \frac{1}{K}\sum_{i=1}^{K} S_{i\rightarrow 0,t}^{gen,to},
\tag{8}
$$

A high value of average total directional connectedness (TCI) reveals an increased risk in the financial market and its low value highlights the low risk. This means shocks in one variable are influenced by its future values and shocks are not transmitted from one variable to another variables.

Finally, the dynamic connectedness approach provides information about net pairwise spillovers relationship between two variables and it is presented in Equation (9):

$$
S_{ij,t}^{gen,net} = gSOT_{ji,t}^{gen,to} - gSOT_{ij,t}^{gen,from}
\tag{9}
$$

If $S_{ij,t}^{gen,net} > 0 \left(S_{ij,t}^{gen,net} < 0\right)$, variable *i* has a higher impact on variable *j* and vice versa, implying that variable *i* dominates variable *j*.

## 4. Empirical Results

In this section we present the results of the dynamic connectedness approach based on the TVP-VAR approach.

### 4.1. The Average Dynamic Connectedness

Table 2 displays the average dynamic connectedness based on the TVP-VAR approach. The diagonal elements in Table 2 are associated with their own contribution to volatility spillover while off-diagonal elements refer to the contribution 'from' or 'to' others. The rows are linked with the contribution of each asset and uncertainty index to forecast error variance of specific asset. Conversely, the columns are associated with the impact of shock in one financial asset to all other assets separately.

**Table 2.** Volatility and return connectedness of cryptocurrency uncertainty indices, cryptocurrencies, precious metals, green bonds, and soft commodities: Evidence using the TVP-VAR approach. **Notes:** TVP-VAR dynamic connectedness approach results with the lag-length of order one by criterion (BIC) with window size of 52 weeks. Panel A includes the dynamic connectedness in full sample, and we tested the dynamic connectedness during COVID-19 in panel B.

| Panel A: Average Dynamic Connectedness Table (Full Sample) | | | | | | | | | |
|---|---|---|---|---|---|---|---|---|---|
| Variables | UCRY Policy | UCRY Price | Bitcoin | Ethereum | Gold | Silver | Platinum | S&P Green Bonds | SP GSCI Softs | FROM |
| UCRY Policy | 43.02 | 47.18 | 1.41 | 0.25 | 0.78 | 2.65 | 2.07 | 1.86 | 0.79 | 56.98 |
| UCRY Price | 35.59 | 58.18 | 0.7 | 0.15 | 0.47 | 1.77 | 1.47 | 1.05 | 0.62 | 41.82 |
| Bitcoin | 8.35 | 14.55 | 44.24 | 4.25 | 2.82 | 6.47 | 5.69 | 10.06 | 3.55 | 55.76 |
| Ethereum | 5.13 | 6.93 | 5.29 | 58.93 | 7.18 | 6.2 | 2.75 | 5.37 | 2.23 | 41.07 |
| Gold | 7.02 | 10.05 | 3.89 | 6.49 | 40.23 | 10.14 | 9.25 | 11.49 | 1.45 | 59.77 |
| Silver | 7.76 | 10.84 | 4.66 | 4.7 | 7.73 | 37.94 | 17.02 | 7.12 | 2.23 | 62.06 |
| Platinum | 15.51 | 25.9 | 3.75 | 1.44 | 5.17 | 12.5 | 27.82 | 6.31 | 1.6 | 72.18 |
| S&P Green Bonds | 6.9 | 9.7 | 7.44 | 1.52 | 3.07 | 5.64 | 5.72 | 54.98 | 5.04 | 45.02 |
| SP GSCI Softs | 7.82 | 13.37 | 4.54 | 2.61 | 1.41 | 4.24 | 9.29 | 12.99 | 43.74 | 56.26 |
| TO | 94.07 | 138.52 | 31.66 | 21.4 | 28.63 | 49.6 | 53.26 | 56.25 | 17.51 | 490.91 |
| Inc.Own | 137.09 | 196.7 | 75.9 | 80.33 | 68.87 | 87.54 | 81.08 | 111.23 | 61.26 | cTCI/TCI |
| NET | 37.09 | 96.7 | −24.1 | −19.67 | −31.13 | −12.46 | −18.92 | 11.23 | −38.74 | 61.36/54.55 |
| NPT | 7 | 8 | 3 | 1 | 1 | 4 | 5 | 6 | 1 | |
| Panel B: COVID-19 Pandemic (1 January 2020 to 31 December 2021) | | | | | | | | | |
| Variables | UCRY Policy | UCRY Price | Bitcoin | Ethereum | Gold | Silver | Platinum | S&P Green Bonds | SP GSCI Softs | FROM |
| UCRY Policy | 47.47 | 35.4 | 0.88 | 3.03 | 1.62 | 0.91 | 3.61 | 2.62 | 4.44 | 52.53 |
| UCRY Price | 39.24 | 45.01 | 0.92 | 2.76 | 1.11 | 0.91 | 3.82 | 2.4 | 3.85 | 54.99 |
| Bitcoin | 5.4 | 6.18 | 18 | 11.6 | 4.5 | 12.28 | 14.38 | 24.81 | 2.85 | 82 |
| Ethereum | 4.6 | 5.41 | 10.45 | 29.06 | 4.19 | 11.65 | 9.57 | 22.61 | 2.46 | 70.94 |
| Gold | 4.05 | 2.5 | 5.12 | 7.32 | 24.81 | 15.83 | 16.42 | 21.19 | 2.76 | 75.19 |
| Silver | 2.28 | 3.19 | 7.78 | 9.06 | 8.91 | 19.39 | 21.61 | 25.52 | 2.26 | 80.61 |
| Platinum | 2.18 | 2.37 | 6.2 | 9.52 | 7.49 | 16.11 | 31.48 | 21.27 | 3.37 | 68.52 |
| S&P Green Bonds | 3.1 | 2.35 | 2.35 | 2.71 | 1.79 | 4.19 | 9.11 | 68.63 | 5.78 | 31.37 |
| SP GSCI Softs | 12.46 | 12.94 | 3.53 | 3.05 | 1.74 | 1.63 | 9.2 | 8.27 | 47.19 | 52.81 |
| TO | 73.3 | 70.34 | 37.24 | 49.04 | 31.36 | 63.52 | 87.71 | 128.69 | 27.77 | 568.97 |
| Inc.Own | 120.77 | 115.35 | 55.24 | 78.1 | 56.16 | 82.91 | 119.2 | 197.31 | 74.95 | cTCI/TCI |
| NET | 20.77 | 15.35 | −44.76 | −21.9 | −43.84 | −17.09 | 19.2 | 97.31 | −25.05 | 71.12/63.22 |
| NPT | 7 | 5 | 2 | 3 | 0 | 3 | 7 | 7 | 2 | |

The findings of the dynamic connectedness network of weekly returns of cryptocurrencies, precious metals, green bonds, soft commodities, and cryptocurrency uncertainty indices display higher internal connectedness with an average total connectedness index (TCI) value of 54.55%. The value of TCI within the dynamic connectedness network explains the higher interconnectedness of these financial assets and cryptocurrency uncertainty indices. The cryptocurrency price uncertainty index transmits the shocks to other assets in the network with a forecast error variance value of 138.52%.

Similarly, during the COVID-19 pandemic, the TCI value is 63.22%, which is higher than full sample results. Moreover, during the pandemic, green bonds with a value of 128.69% transmit the shock to other assets and uncertainty indices. The connectedness is

increased during the COVID-19 pandemic. The increase in TCI during COVID-19 pandemic suggest that volatility spillover and transmission of risk increase during financial turmoil. These findings are consistent with previous research (Akhtaruzzaman et al. 2021; Boubaker et al. 2016; Bouri et al. 2021b; Costa et al. 2022), as they also find that total connectedness is increased during crisis times.

Overall, these findings strongly suggest that the selected financial assets are closely connected, and shocks are transmitted from cryptocurrency uncertainty indices to other financial assets. Hence, risk-averse investors should take these findings into account for investing in these financial assets, and they can diversify their portfolios by investing in assets with low interconnectedness. Investors should consider cryptocurrency uncertainty before investing in these financial assets, especially during the COVID-19 pandemic.

### 4.2. The Dynamic Total Connectedness

It is important to note that, in Table 2, results about dynamic connectedness across time are not included. Moreover, we cannot see the connectedness during the Global Financial Crisis (GFC), COVID-19 pandemic, and other influential and/or extreme events. Figure 2 illustrates the dynamic total connectedness (TCI) across time to explain the volatility transmission across financial assets. As shown in Figure 2, the total connectedness is within the 45% to 95% range. The TCI is higher during 2016 and it remained at 55% from 2018 to the first days of 2020. However, the TCI values sharply increase during the first wave of the COVID-19 pandemic, with a TCI value of around 77%.

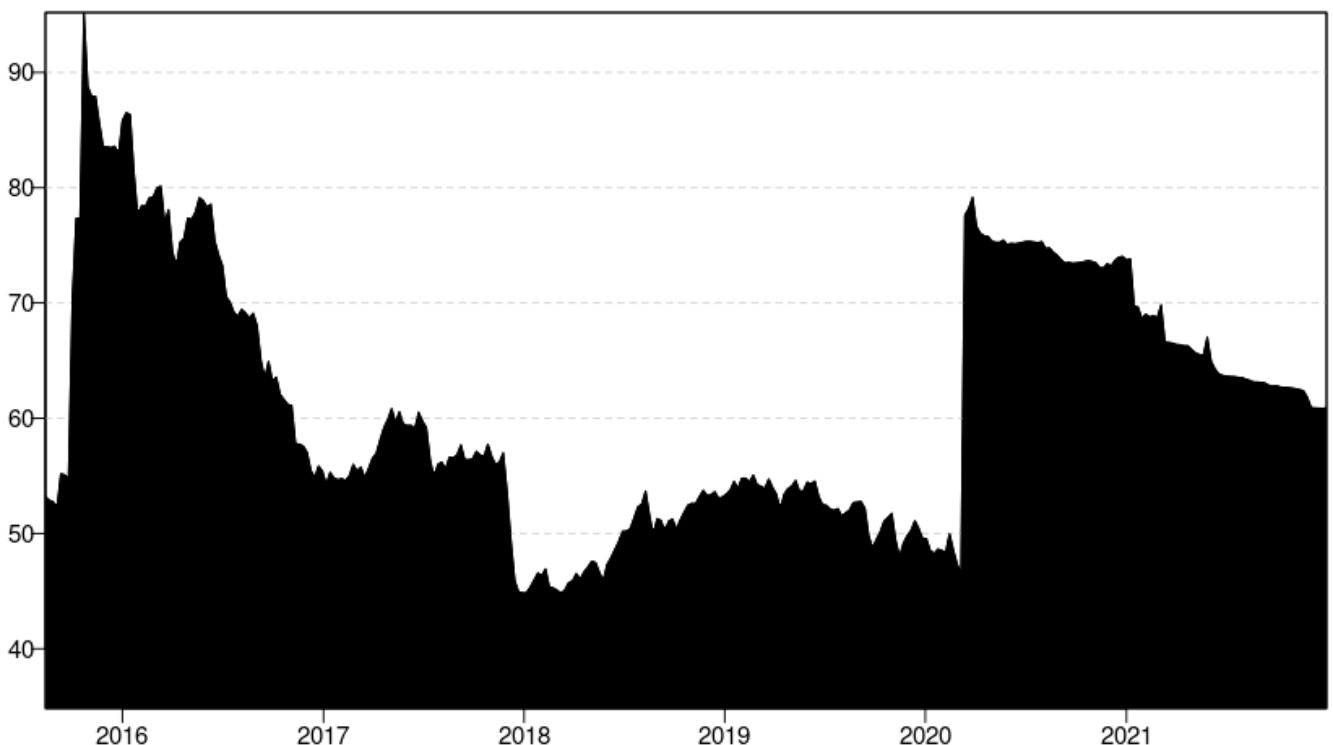

**Figure 2.** Dynamic total connectedness. **Notes:** Figure 2 shows the results of the TVP-VAR approach with lag-length of order one by BIC criterion and a 20 step-ahead generalized forecast error variance decomposition (FEVD). The black area in the figure represents the total connectedness (TCI).

The consistent presence of the TCI value above 50% across a majority of time frames provides substantial evidence for heightened return and volatility spillover within these financial assets. This trend signifies that elevated cryptocurrency uncertainty indices wield a notable influence over these financial components. The amplified dynamic connectedness values recorded during the COVID-19 pandemic can be attributed to the escalated apprehension among investors triggered by negative news related to the COVID-19 situation. Concurrently,

the surge in economic policy uncertainty further compounds this situation. It is noteworthy that cryptocurrencies were particularly susceptible to changes in the COVID-19 pandemic scenario (Allen 2022; Lahmiri and Bekiros 2020; Salisu and Vo 2020).

### 4.3. Net Total Directional Connectedness

Net total directional connectedness results are displayed in Figure 3 This figure presents the time-varying role of net receiving or net transmitting role of financial assets. Figure 4 shows that the UCRY policy and price indices remain stable before the COVID-19 pandemic and that the UCRY price index acts as a transmitter of shocks during the first days of COVID-19; these findings are consistent with Lundgren et al. (2018). UCRY policy index is a net receiver of the shocks. As shown in Figure 3, Ethereum, Bitcoin, gold, and soft commodities are net receivers of shocks. The silver, platinum, and green bonds are net receivers of shocks before COVID-19, but they become the net transmitter of shocks during it. Overall, most of the assets are net receivers of shocks and their spillover behavior changes during the pandemic.

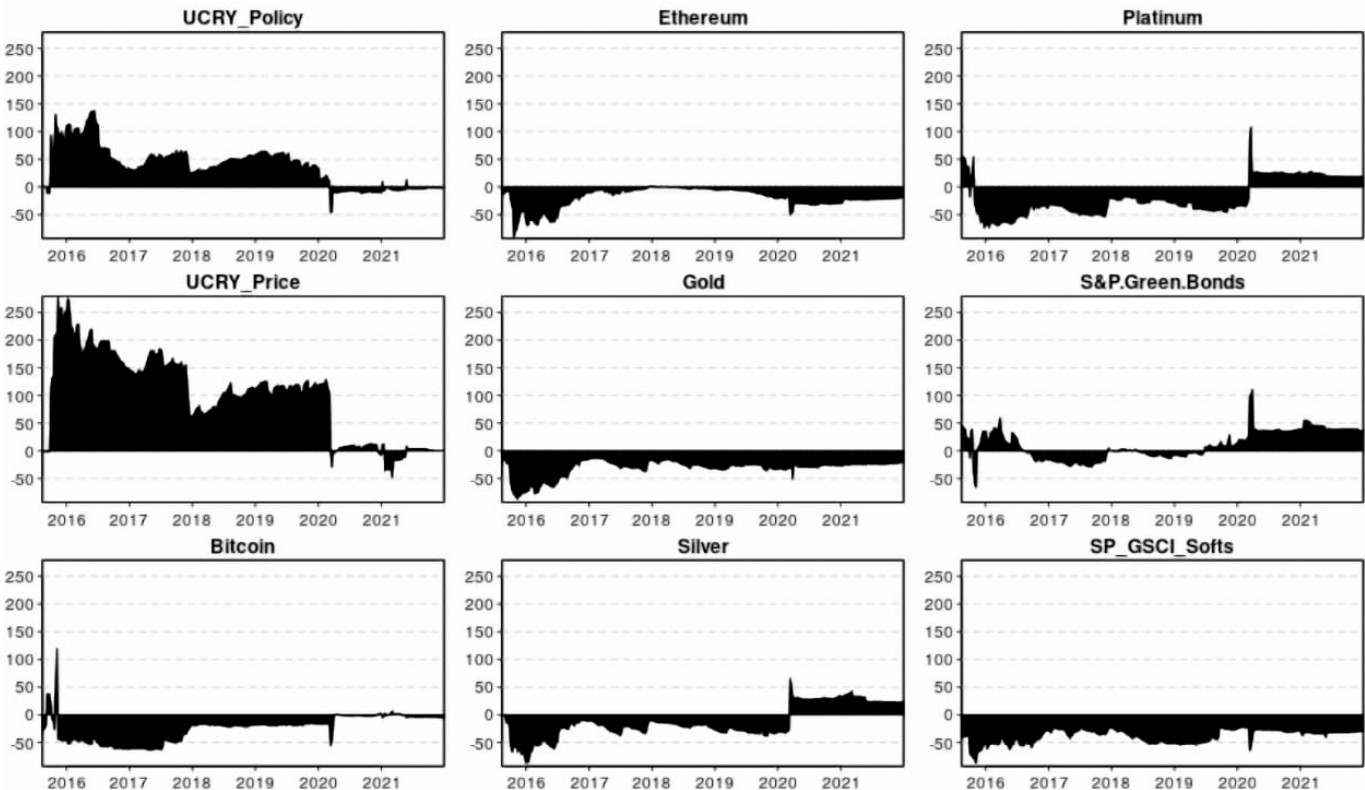

**Figure 3.** Dynamic net total directional connectedness. **Notes:** Figure 3 shows net total directional connectedness based on the TVP-VAR approach with lag-length of order one by BIC criterion and a 20 step-ahead generalized forecast error variance decomposition (FEVD). The black area represents the net total directional connectedness. Meanwhile the positive values show a net transmitter role, and the negative values indicate the net receiving role of financial assets.

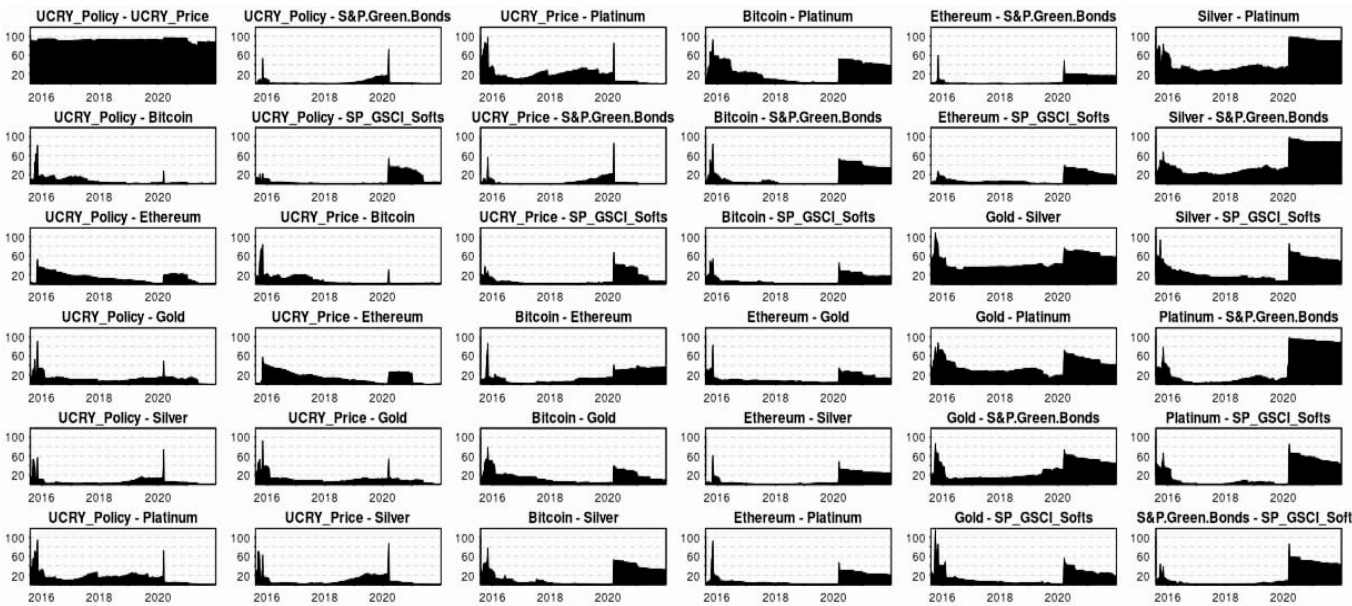

**Figure 4.** Dynamic net pairwise total directional connectedness. **Notes:** Figure 4 shows net pairwise total directional connectedness based on the TVP-VAR approach with lag-length of order one by BIC criterion and a 20- step-ahead generalized forecast error variance decomposition (FEVD). The black area represents the net total directional connectedness.

### 4.4. Net Pairwise Directional Connectedness

The results of time-varying net pairwise connectedness between cryptocurrency uncertainty indices and financial assets are presented below in Figure 4. The pairwise connectedness is higher between UCRY indices, cryptocurrencies, and precious metals, especially during the COVID-19 pandemic. The connectedness between precious metals, soft commodities, and green bonds is also higher, especially during the outbreak's spread. Overall, the magnitude of net pairwise connectedness is higher, and the spillover patterns were changed during the COVID-19 pandemic, which is consistent with findings of other studies (Bouri et al. 2021a; Elsayed et al. 2022a; Le et al. 2021a). These findings suggest that investors should consider persistence of asset before investing as patterns of spillovers were changed during COVID-19 pandemic.

### 4.5. Dynamic Connectedness Network Plot

Figure 5 illustrates the network plot of the return and volatility connectedness between cryptocurrency uncertainty indices and different financial assets. The UCRY price and policy indices are net transmitters of shocks to Bitcoin, Ethereum, gold, silver, platinum, and soft commodities. Moreover, the green bonds are net transmitter of shocks towards gold and soft commodities. The net total directional connectedness between UCRY price to Bitcoin, Platinum, and gold is higher because the node size is large. Our findings suggest that equity market investors should look for volatility spillovers from cryptocurrency uncertainty indices towards different financial assets before investing in these assets during a financial crisis. These findings suggest that investors in traditional markets should be cautious during financial turmoil and its influence on traditional assets as our findings show that cryptocurrency uncertainties transmit the shocks towards traditional assets market; hence, traditional investors experienced lower returns during the COVID-19 pandemic.

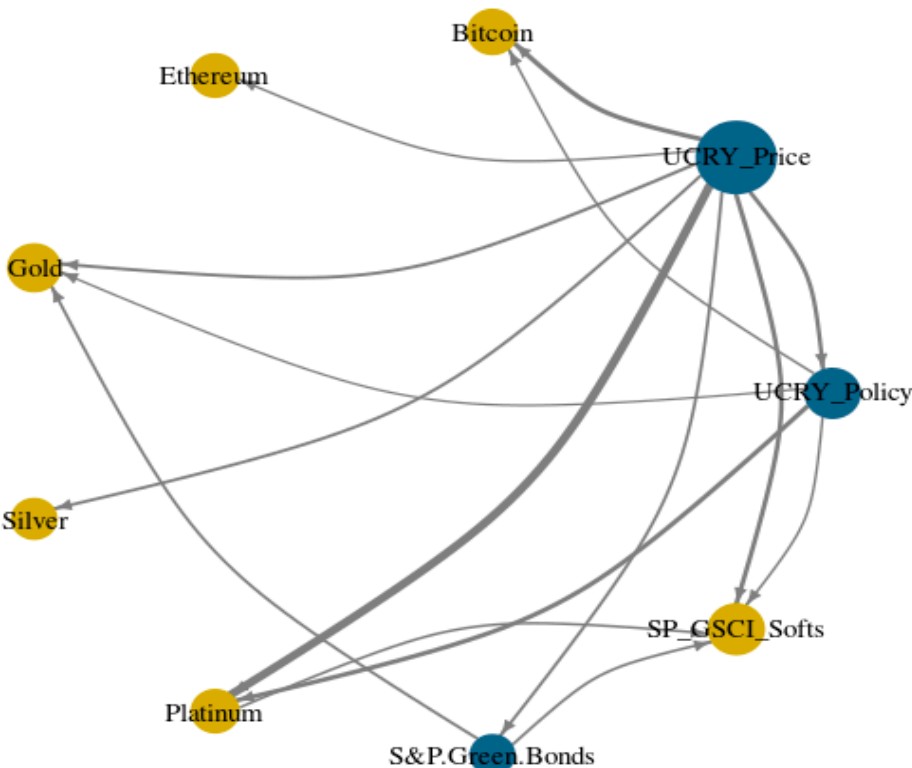

**Figure 5.** Dynamic connectedness network plot. Notes: Figure 5 displays the network plot using the TVP-VAR approach. The blue nodes represent the net transmitter role while the yellow nodes illustrate the net receiver of shocks. The node sizes show the weighted average net total directional connectedness.

## 5. Concluding Remarks

This study examines the dynamic connectedness of return and volatility spillover among cryptocurrency uncertainty, cryptocurrencies, green bonds, precious metals, and soft commodities. The investigation relies on weekly returns data from 7 August 2015 to 31 December 2021, using the TVP-VAR approach as detailed by Antonakakis et al. (2020). The total connectedness is higher, a trend particularly heightened during the COVID-19 pandemic. During this crisis, the cryptocurrency policy uncertainty index emerged as the primary transmitter of shocks to other financial assets, while the cryptocurrency price index assumed the role of shock receiver of shock during COVID-19. The pandemic has instigated shifts in returns and volatility connectedness across these financial assets. For instance, certain assets that were previously net shocks receivers transitioned into shock transmitters during the COVID-19 outbreak. Moreover, the pandemic has fostered heightened connection among precious metals, soft commodities, and green bonds. Precious metals and cryptocurrencies, as recipients of shocks, warrant particular attention from investors and practitioners who can opt for alternative assets as a strategy to hedge the cryptocurrency uncertainty and reduce the portfolio risk in times of financial turmoil.

These findings hold considerable implications, urging investors to carefully assess volatility spillovers from cryptocurrency uncertainty indices into traditional markets for comprehensive diversification insights across assets. Consequently, policymakers and investors are encouraged to scrutinize cryptocurrency uncertainty spillover patterns onto various traditional markets, enabling them to optimize returns through diversified global asset portfolios—especially crucial amidst financial disturbances.

To acknowledge this study's limitations, it is worth noting that the availability of UCRY uncertainty indices in a weekly frequency prompted the utilization of weekly data. However, this choice may omit some critical information. For future investigations, re-

searchers should consider constructing a daily cryptocurrency uncertainty index to explore its dynamic connectedness with other assets, thereby offering a more nuanced perspective.

**Author Contributions:** Conceptualization, M.A.; Methodology, A.A., and M.A.; Software, A.A.; Validation, A.A., and M.A.; Formal analysis, A.A.; Investigation, A.A.; Resources, M.A.; Data curation, M.A.; Writing—original draft preparation, A.A., and M.A.; Writing—review and editing, A.A., M.I.B., and M.A.; Visualization, A.A., M.I.B., and M.A.; Supervision, M.I.B.; Project administration, M.I.B. All authors have read and agreed to the published version of the manuscript.

**Funding:** This research received no external funding.

**Data Availability Statement:** The data presented in this study are openly available in the cryptocurrency uncertainty index, the coinmarektcap.com, and thomsonreuters.com.

**Conflicts of Interest:** The authors declare no conflict of interest.

## Notes

1.  The cryptocurrency uncertainty index. Finance Research Letters, 102147. The latest UCRY Weekly Index data can be downloaded from: https://sites.google.com/view/cryptocurrency-indices/the-indices/crypto-uncertainty?authuser=0 (accessed on 8 January 2022).
2.  https://coinmarketcap.com/ (accessed 8 January 2022).
3.  https://www.thomsonreuters.com/en.html (accessed on 8 January 2022).

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
