# Peer review of "Interconnectedness of Cryptocurrency Uncertainty Indices with Returns and Volatility in Financial Assets during COVID-19"

_jrfm, doi:10.3390/jrfm16100428_

Round 1

Reviewer 1 Report

Dear authors,

Thank you for the opportunity to review this manuscript.

This paper applies the time-varying parameter vector auto-regressions (TVP-VAR) aims to investigate the dynamic relationship between cryptocurrency uncertainty indices and the movement of returns and volatility in various financial assets (cryptocurrencies, precious metals, green bonds, and soft commodities). I believe the article may be of interest to some readers, however it needs to be improved in some respects for publication.

1. Introduction. This section is well-balanced, it introduces the reader into the topic of this research paper. The research questions are interesting.

However, the authors should conduct a more detailed literature review on previous papers that has dealt the role of cryptocurrency uncertainty indices, highlighting gaps in the literature and their real contributions to these issues.

2. Literature Review. I particularly appreciate the fact that the authors carefully review the specialized literature in this particular subfield, using recent references from top journals.

I suggest the addition of references that could help improve you work:

Asl, M. G., Canarella, G., & Miller, S. M. (2021). Dynamic asymmetric optimal portfolio allocation between energy stocks and energy commodities: Evidence from clean energy and oil and gas companies. Resources Policy, 71, 101982.

Iglesias-Casal, A., López-Penabad, M. C., López-Andión, C., & Maside-Sanfiz, J. M. (2020). Diversification and optimal hedges for socially responsible investment in Brazil. Economic Modelling, 85, 106-118.

Shahid, M. N., Azmi, W., Ali, M., Islam, M. U., & Rizvi, S. A. R. (2023). Uncovering risk transmission between socially responsible investments, alternative energy investments and the implied volatility of major commodities. Energy Economics, 120, 106634.

3. Research Methods. I appreciate the way the methodology is clearly described and the equations are presented.

4. Empirical results. I believe the methodology used by the authors is appropriate and the steps are correctly taken.

I appreciate Tables 2 and Figures 2, 3, 4, 5 and then the Econometric analysis, too.

The explanation of the results has to be connected to the literature and also explained in depth from an economic perspective. This version does not discuss the practical implications of the empirical results. You should try to improve this part by answering questions such as why this result, why this behaviour of the dynamic correlation, what is the mechanism leading to this movement, and/or the financial implications (e.g. implications for portfolio optimisation, asset allocation, etc.)

5. Conclusions.

While the conclusions are partly well drawn, they can be improved.

As noted, one shortcoming of this article that could be improved is its lack of a clear explanation of how its findings could be beneficial to market participants and regulators, among others.

Also, I think the authors should emphasize the limits of their paper.

Minor aspects:

Citations of references in the text should be reviewed. On certain occasions, they are cited by first name and not by surname, as stated in the bibliography.

Some references are incomplete. You should check the form of citation in the text.

I hope you find my comments helpful to improve your paper. Good luck.

Author Response

Pls see enclose copy of both reviewers

Reviewer 2 Report

Detailed notes attached.

Author Response

Reply to reviewer 2 report is on pages 3 & 4 at the end of this document.
